# Pairing a high-resolution statistical potential with a nucleobase-centric sampling algorithm for improving RNA model refinement

Peng Xiong[1,2], Ruibo Wu[3], Jian Zhan [1✉] & Yaoqi Zhou [1,2,4✉]

Refining modelled structures to approach experimental accuracy is one of the most challenging problems in molecular biology. Despite many years' efforts, the progress in protein or RNA structure refinement has been slow because the global minimum given by the energy scores is not at the experimentally determined "native" structure. Here, we propose a fully knowledge-based energy function that captures the full orientation dependence of base–base, base–oxygen and oxygen–oxygen interactions with the RNA backbone modelled by rotameric states and internal energies. A total of 4000 quantum-mechanical calculations were performed to reweight base–base statistical potentials for minimizing possible effects of indirect interactions. The resulting BRiQ knowledge-based potential, equipped with a nucleobase-centric sampling algorithm, provides a robust improvement in refining near-native RNA models generated by a wide variety of modelling techniques.

[1] Institute for Glycomics and School of Information and Communication Technology, Griffith University, Parklands Drive, Southport, QLD, Australia. [2] Institute for Systems and Physical Biology, Shenzhen Bay Laboratory, Shenzhen, China. [3] School of Pharmaceutical Sciences, Sun Yat-Sen University, Guangzhou, China. [4] Peking University Shenzhen Graduate School, Shenzhen, P. R. China. ✉email: j.zhan@griffith.edu.au; zhouyq@szbl.ac.cn

Three-dimensional structures of RNAs offer the best clues for their functions and yet only a few thousand structures have been determined[1], compared to 24 million of non-coding RNA sequences collected in RNAcentral (as of October 25 2020)[2]. This huge gap continues to expand exponentially because sequencing entire transcriptomes is only a fraction of the cost and time required for structure determination of a single RNA by the techniques such as X-ray crystallography, nucleic magnetic resonance, and cryo-electron microscopy.

One possible solution to this growing problem is to predict RNA structures from their sequences by computational template-based homology modeling and de novo folding[3,4]. Homology modeling such as RNABuilder[5] and ModeRNA[6] attempts to predict the structure of a target RNA by mapping the query sequence onto the template structure(s) of its homologs. However, most RNAs do not have their respective homologous templates. Thus, method development was mainly devoted to predicting structures by de novo folding from sequences through fragment or motif assembling. Examples are FARNA[7], MC-fold/MC-Sym[8], Rosetta-SWM[9], ifoldRNA[10,11], Vfold[12,13], 3dRNA[14,15], and SimRNA[16]. Evaluation of these methods by RNApuzzles[17–19] indicates that reasonable predictions are possible only for RNAs with either simple topology or existing homologous templates. Moreover, reasonable near-native predictions were often ranked poorly by the predictors. Clearly both homology and de novo models would greatly benefit from a structure-refinement technique that can bring the models closer to native structures locally and globally and improve near-native ranking.

Structure refinement, however, is challenging with few methods available for RNAs. Even for protein structure refinement, despite a long history of method development, only a moderate local improvement can be made by combining molecular mechanics force fields and knowledge-based potentials with pre-defined restraints to prevent large deviation away from the native basin[20,21]. For RNAs, most all-atom refinements were performed after coarse-grained sampling for removing steric clashes and non-ideal bond lengths, bond angles, or torsional angles. For example, the QRNAS refinement[22] utilizes a modified AMBER force field with enforced base-pair planarity, explicit hydrogen bonds, and backbone regularization. FARFAR[23,24] improves model accuracy through all-atom refinement after FARNA coarse-grained sampling. It employed a Rosetta all-atom force field that mixes physical and knowledge-based scores with RNA-specific terms for Watson–Crick (WC) base pairing, base stacking, and torsional potentials[25].

Here, we build a knowledge-centric refinement energy score. Atomic-level knowledge-based energy functions, derived from known three-dimensional structures, have traditionally focused on distance dependence in protein structures[26] with similar statistical potentials developed for RNA[27–29]. Unlike these atomic knowledge-based scores and previous all-atom RNA refinement energy scores[22,23], the statistical potential in this work is tailored specifically to RNA interactions that were dominant by orientation-dependent base-pairing and stacking interactions[30] with rotameric backbone[31], in contrast to dominant distance-dependent hydrophobic interactions[32] with rotameric sidechains[33] in protein folding. Base–base interactions were described by fine grids in a six-dimensional orientational space and scaled by quantum mechanical calculations allowing better capture of both local and global interactions. This Backbone Rotameric and Quantum-mechanical-energy-scaled base–base knowledge-based potential (BRiQ) is integrated with a new nucleobase-centric tree algorithm that samples backbone conformations around predicted or known base pairs. The resulting refinement technique can consistently improve model structures for the majority of near-native structural models as demonstrated

by refining Rosetta-SWM motif[34], RNA puzzles[17–19], and FARFAR2[24] models.

## Results

**BRiQ refinement energy score.** The BRiQ refinement energy score is designed to capture stably stacked bases linked by a more flexible ribose and phosphate backbone. In particular, the interactions associated with bases and oxygen atoms in backbones are strongly orientation-dependent. To capture this unique structural property of RNA chains, the energy score ($E$) is separated into six terms: orientation-dependent interactions between two bases ($E_{bb}$), between a base and a main-chain oxygen atom ($E_{bo}$), and between two main-chain oxygen atoms ($E_{oo}$), backbone rotameric energy ($E_{rot}$), internal energy ($E_{internal}$), and atomic clash energy ($E_{clash}$). That is,

$$E = E_{bb} + E_{bo} + E_{oo} + E_{rot} + E_{internal} + E_{clash} \qquad (1)$$

where we employed five oxygen atom types including OP for OP1 and OP2 and O2', O3', O4', and O5' in ribose. To capture the orientation dependence, the whole nucleobase (A, C, G, and U) is treated as a single rigid group with a local coordinate system and the relative position between two bases is described by the distance vectors and their orientations (Fig. 1A). The densities in the six-dimensional space were derived from known RNA structures using kernel density estimations, rather than histogram statistics (see Methods). This allows a smoother energy landscape (Fig. 1B) and the near-native local minima are closer to the corresponding native structure in a fine orientational space. To remove the impact of indirect base–base interactions, statistical energy scores are scaled by quantum mechanical calculations according to the correlation coefficient between representative hydrogen-bonded base pairs (Fig. 1B, C). Similarly, kernel density estimations were also employed to obtain base–oxygen (Fig. 1D) and oxygen–oxygen interactions whereas statistical rotameric states were obtained for ribose backbones (Fig. 1E). Empirical internal bond angle, bond length, and atom-crash energies were also developed to ensure appropriate bond geometry and packing density (Methods).

**NuTree sampling algorithm.** The above nucleobase-centric energy score is coupled with a nucleobase-centric tree (NuTree) algorithm for conformational sampling, in which each node is a base and each edge represents the relative position between two bases (Fig. 1F). Two connected bases in the folding tree could be sequential neighbors or hydrogen-bonded base pairs. Instead of sampling in backbone-dihedral angle space, conformations are sampled according to the pre-defined orientation space of each edge type because the orientations around more stable bases are the determinants of backbone orientations. For the same reason, coordinates of sidechains were built prior to those of the backbones. Only local moves were allowed for each node whereas both local and global moves were allowed for each edge, making both local and global structural improvements possible. In the process of Monte Carlo sampling, we only need to calculate the energy change upon a random structure move, without knowing the first or second derivative of the energy function. Therefore, this sampling algorithm could be used for optimizing an RNA structure with a non-differentiable energy function in atomic accuracy.

**Refining Rosetta-SWM conformations for RNA motifs.** We first test our method using a benchmark of 48 motifs. This benchmark[34] was developed to examine the ability of Rosetta-SWM to recover the loop region given some of the base pairs and environment contacts fixed (See Methods and Supplementary

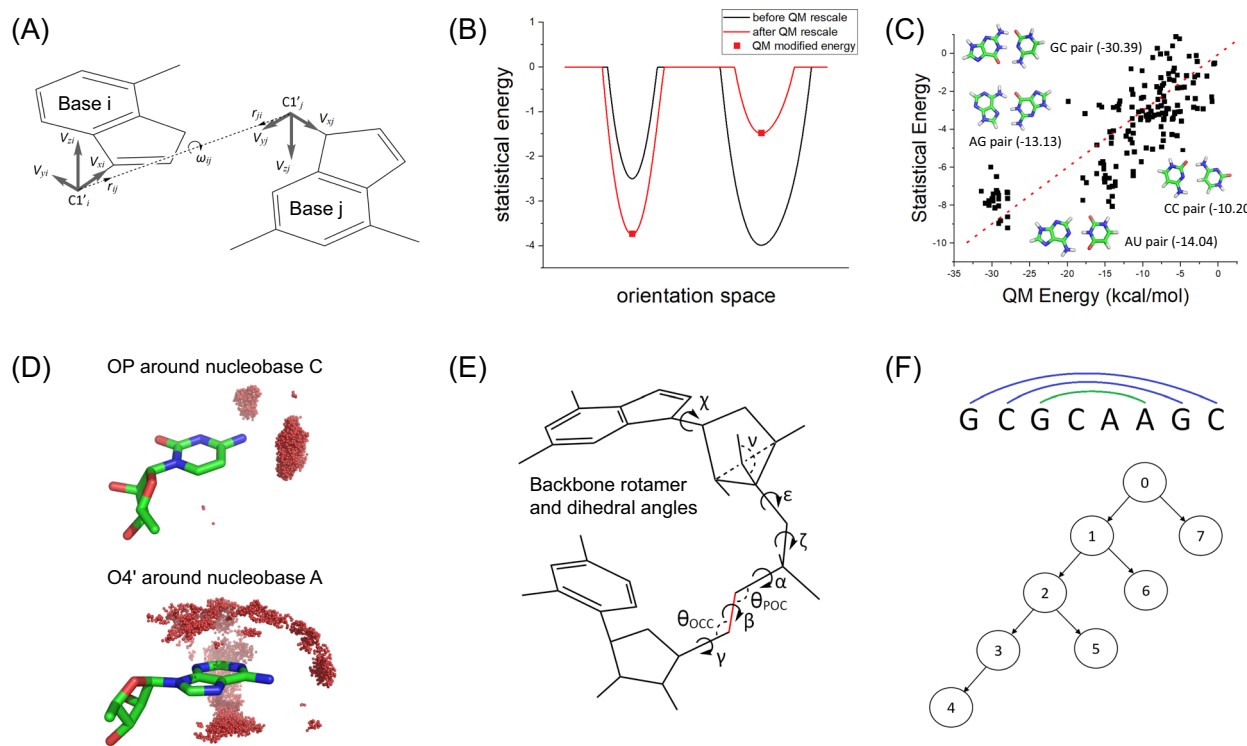

**Fig. 1 Nucleobase-centric statistical potential and sampling technique for RNA structure refinement. A** Six-dimensional base–base statistical potentials with relative positions defined by $\mathbf{r}_{ij}$ and $\mathbf{r}_{ji}$ vectors along with the rotational angle $\omega_{ij}$. **B** A schematic illustration of the orientation dependence before and after Quantum Mechanical (QM) energy scaling as labeled. **C** This QM scaling is based on the correlation between the statistical energy scores of hydrogen-bonded base pairs ($-\ln P_{hb}$) and QM calculations. $P_{hb}$ is the probability of hydrogen-bonded base pairs. The QM energies for a few base pairs were illustrated in the insert. **D** The distribution of OP (red dots) around nucleobase C and the distribution of O4′(red dots) around the nucleobase A as labeled. **E** Backbone rotamers defined according to various torsion and improper angles that control the ribose ($\chi, \nu$) and phosphate ($\varepsilon, \zeta$) backbone. Dihedral angles $\alpha$, $\beta$, and $\gamma$, bond angles $\theta_{POC}$, $\theta_{OCC}$ and bond length of C5′O5′ were required to calculate the internal energy. **F** Nucleobase-centric fold-tree (NuTree) algorithm for refinement by defining bases as the nodes and locally or globally connected bases as the edges, illustrated by the GCAA tetraloop with the canonical and noncanonical base pairs shown in blue and green colors, respectively.

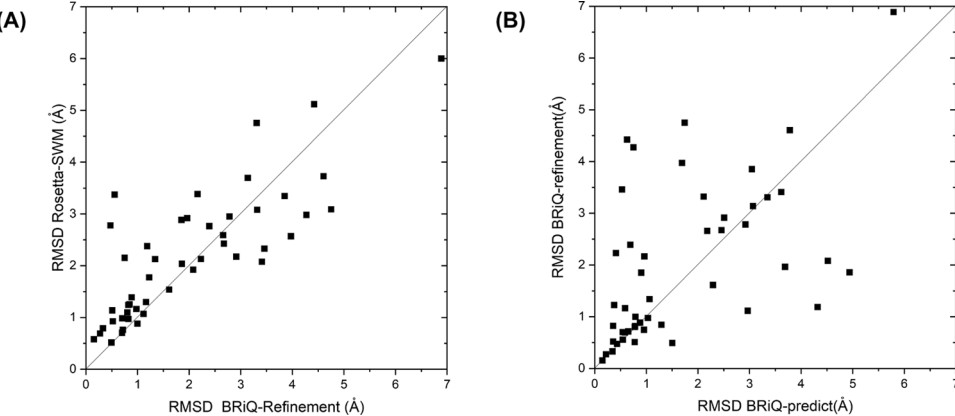

**Fig. 2 Refinement of Rosetta-SWM models and base-pair-restrained prediction of RNA structural motif by BRiQ. A** Refining Rosetta-SWM motif models by BRiQ improves the majority of those model structures with RMSD<2 Å as demonstrated by RMSD comparison (lowest RMSD of top 1% before refinement on Y-axis versus after refinement on X-axis). **B** Fixing all native base pairs with random initial conformations for all other regions and then folding the remaining structure leads to more accurate motif models for the majority of the motifs than refining Rosetta-SWM models that have pre-assigned, partially fixed base pairs (Lowest RMSD of top 1% BRiQ-refined models on Y-axis versus lowest RMSD of top 1% BRiQ models with fixed native base pairs on X-axis).

Data 1). We test our refinement method by refining all 88,352 conformations generated by Rosetta-SWM with the NuTree algorithm and the BRiQ energy score. The NuTree was constructed with the base-pairing information extracted from Rosetta-SWM model, and the BRiQ energy was optimized by Monte Carlo sampling at low temperature. The lowest RMSD values of top 1% predicted models by the Rosetta and BRiQ energies are compared in Fig. 2A. The majority of predicted motif conformations (31/48, 64.5%) have an improved RMSD after BRiQ refinement with a median reduction of 0.2 Å RMSD change

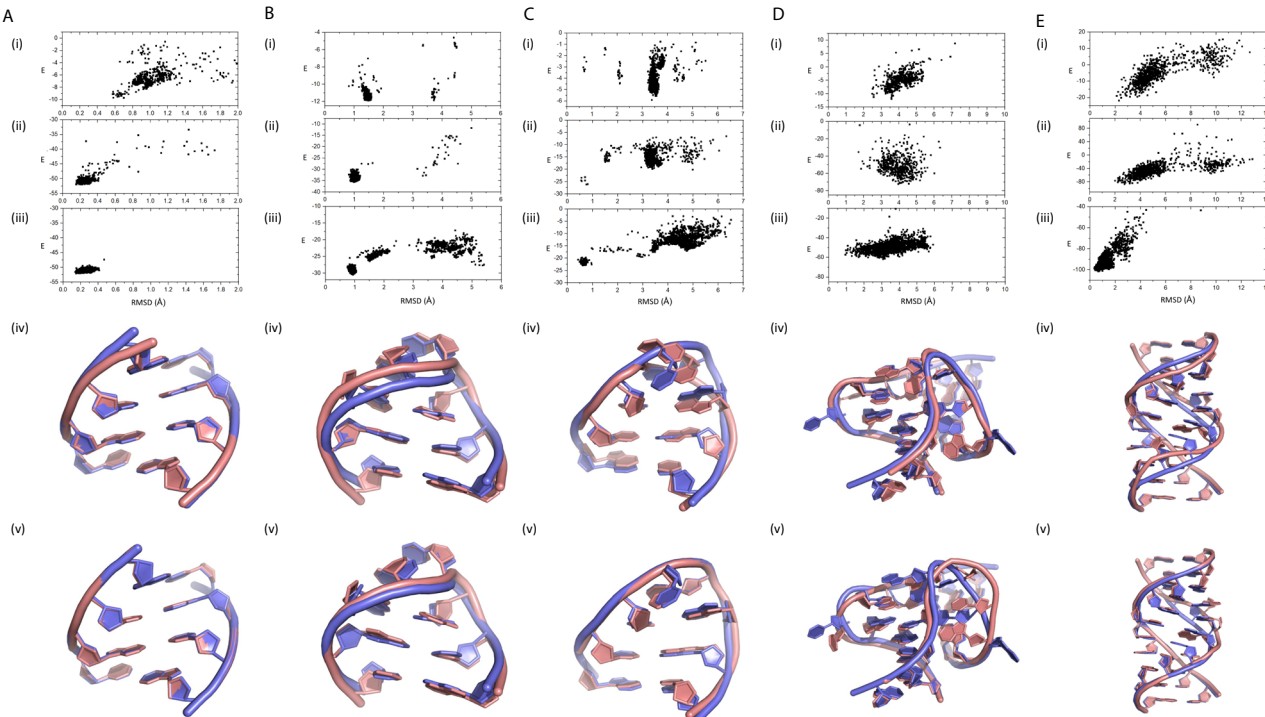

**Fig. 3 Comparison of representative motif models from Rosetta-SWM, refinement of Rosetta-SWM models and base-pair-restrained prediction by BRiQ.** Energy versus RMSD values of the conformations sampled by (i) Rosetta-SWM, (ii) refinement of Rosetta-SWM conformations by BRiQ, (iii) structure prediction by BRiQ with all native base pairs fixed for **A** CG-helix, **B** GCAA-tetra loop, **C** UUCG-tetra loop, **D** j55a-P4P6-fixed, and **E** loopE-fixed. Structure alignment of native (blue) to the best in top 1% predicted models by Rosetta-SWM (panel iv) and BRiQ refinement (panel v) in the bottom panel.

among all 48 motifs. The improvement is more impressive on the refinement from near-native models (RMSD of Rosetta-SWM model less than 2 Å), 81% (17/21) predicted conformations have an improved RMSD after BRiQ refinement. The RMSDs of the top 1% predicted models for the remaining four motifs (4/21) are essentially unchanged.

**Generate motif structures with base-pairing information**. As the NuTree sampling algorithm is a refinement protocol, near-native conformations will be difficult to obtain if initial conformations contain false base pairs generated by other methods. Here, we further test whether or not near-native structures can be generated from the NuTree algorithm if all motif base pairs are known. Starting from a random backbone conformation with base pairs fixed, we run Monte Carlo sampling at different temperatures (Methods) to optimize the BRiQ energy score. The lowest RMSD of top 1% predicted models by the BRiQ score are compared in Fig. 2B. The majority of predicted conformations (33/48, 68.8%) have an improved RMSD with correct and complete base-pairing information, compared to partial fixation by Rosetta-SWM. The median improvement in the best RMSD value within top 1% predicted models is 0.29 Å.

Five representative motifs were chosen for illustration in Fig. 3. They are a standard helix made of GC pairs (Fig. 3A), two tetraloops (GCAA-tetra loop, Fig. 3B, UUCG-tetra loop, Fig. 3C) commonly used for testing molecular mechanics force fields[35], one example of refinement leading to worse structures (j55a-P4P6-fixed, Fig. 3D) and an example of internal loop (loopE, Fig. 3F). In these figures, the top to bottom panels represent the results from Rosetta-SWM, refinement of Rosetta-SWM conformations by BRiQ and the structure prediction by BRiQ with native base-pairing information. For the GC helix (Fig. 3A), minimum RMSD sampled by Rosetta-SWM is 0.58 Å, compared to 0.14 Å by BRiQ. For GCAA-tetra loop (Fig. 3B), near-native

local minimal improved from around 1.5 Å RMSD by Rosetta-SWM to about 1 Å by BRiQ. UUCG-tetraloop (Fig. 3C) is an example that <1 Å near-native structure is the global minimum in BRiQ, whereas the lowest energy conformation in Rosetta-SWM is around 3.5 Å. For these three motifs, BRiQ refinement of Rosetta-SWM models leads to an improved backbone fitness to the native backbone. Motif j55a-P4P6-fixed (Fig. 3D) is an example of worse RMSD values after refining Rosetta-SWM conformations. We noticed that this native structure has one base protruded into the solvent, which may be the reason for the failure of BRiQ as the solvation effect is only implicitly accounted for in a statistical potential. For loopE, the best predicted conformation by Rosetta-SWM is about 2 Å RMSD (panel Ei). However, these models have an incorrectly folded non-WC pair that prevented BRiQ to make significant further improvement over Rosetta-SWM (panel Eii). If native non-WC pairs were employed, we would obtain 0.4 Å RMSD for the best within top 1% (panel Eiii). We can achieve this high-resolution structure even without using any non-WC pairs as restraints. We further found that if native conformations are directly refined by BRiQ, the lowest energy conformations of the most motifs (45/48, 94%) remain <2.5 Å away from the native structure. Refining models from the native structures and the Rosetta-SWM models finds the same minimum for most motifs with similar energy. As a result, most motifs have high-quality near-native conformations as the global minimum.

**RNA puzzle refinement**. A real-world test of any refinement techniques is to refine previously predicted models. Here we refined all submitted models in 24 RNA puzzle experiments (PZ1-PZ25, Supplementary Data 2). For each submitted model, we generated 20 refinement models. Then, the lowest energy model within 20 refinement models is treated as the BRiQ predicted model. The lowest RMSD model from all BRiQ predicted

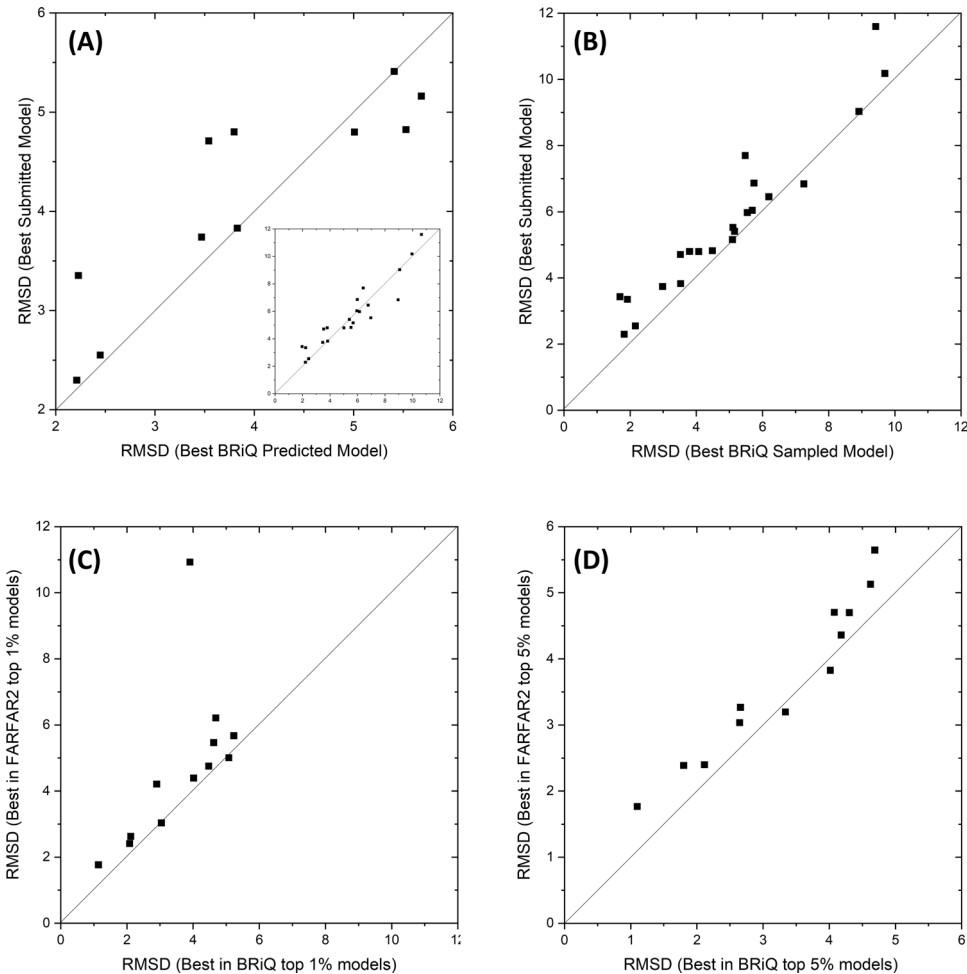

**Fig. 4 Refinement of RNA puzzle models and FARFAR2 models by BRiQ.** BRiQ refinement achieves consistent improvement over the best-submitted models for RNA puzzles containing model structures with RMSD <4 Å (**A**), over all RNA puzzles in conformational sampling (**B**), over FARFAR2 models for 10 out of 12 RNA puzzles in top 1% (**C**) and top 5% (**D**) predicted models.

models for a given RNA puzzle is compared to the lowest RMSD value among all submitted models in Fig. 4A. As the figure shown, if initial models contain models with RMSD < 4 Å, refinement always leads to improvement, with the largest RMSD reduction of 1.5 Å. Figure 4B further compares the lowest RMSD value among all submitted models to those lowest RMSD values among all BRiQ refined models. Essentially all RNAs have better models sampled after refinement with the largest RMSD reduction at 2.2 Å.

Another metric for describing structural difference is deformation index (DI) proposed by Parisien et al.[36], which emphasized on similarity in the base–base interaction network. Supplementary Data 2 also showed the results based on DI. Indeed, we found that more RNAs showed the improvement in DI after refinement than in RMSD. For example, 20/25 predicted DI values in RNA puzzles were reduced, compared to only 13/25 in predicted RMSD.

MolProbity[37] is a tool for checking the quality of RNA structures. We calculated MolProbity scores for all RNApuzzle models. Supplementary Fig. S1 shows that the clash scores of 75 or larger are all decreased to less than 50 after BRiQ refinement. The average clash score reduced 40% from 20.87 to 12.58. Except a few outliers, the clash scores are less than 30 after refinement.

In addition to the whole motif, we further analyzed the refinement results of RNA puzzles at the base-pair level. Here, we employed DDM to measure the relative orientational difference

between predicted and native base pairing structures according to four pseudo atoms employed for representing each base (see Methods). As Supplementary Data 3 shows, the average DDM values from native base-pairing structures of Watson–Crick pairs, non-Watson–Crick pairs, and base-stacking decreased 30% from 0.545 to 0.384, 17% from 0.687 to 0.570, and 22% from 0.834 to 0.650, respectively. The improvement in base pairing structures after BRiQ refinement is found for essentially all RNA puzzles (except for non-Watson–Crick pairs in PZ02, PZ03, and PZ05), regardless whether or not there is an improvement of the overall RMSD or not. More improvement at the base-pair level indicates that the BRiQ refinement occurred at the detailed atomic resolution.

It is interesting to know what happens if BRiQ is applied to refining native structures. We examined the deviation from the native structure at the base-pair structural level. Supplementary Fig. S2 shows the change of base pair structures in DDM as a function of X-ray structure resolution after refinement of native structures by BRiQ. Overall changes to the native base pairing structures are small. There is a trend that larger changes in base pair conformations were observed for lower resolution structures, suggesting more uncertainty for low resolution structures as expected.

**Comparison to FARFAR2**. We also test our refinement method starting from the recently developed FARFAR2 predicted

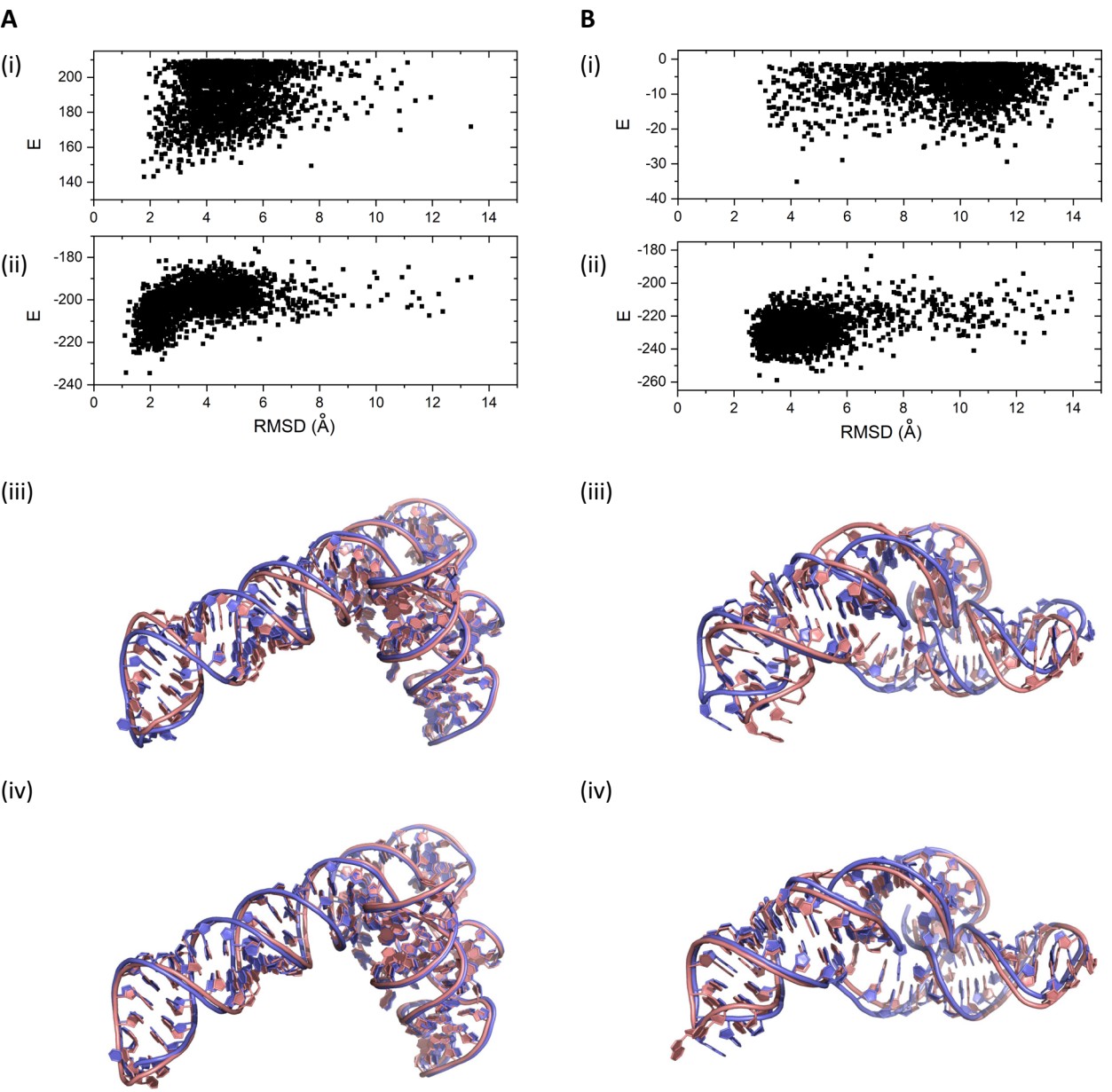

**Fig. 5 Comparison of representative FARFAR2 models and their refinement by BRiQ.** Energy versus RMSD values of the conformations sampled by (i) FARFAR2, (ii) refinement of FARFAR2 conformations by BRiQ for **A** RNA Puzzle 4 and **B** RNA Puzzle 18. Structure alignment of native (blue) to the best in top 1% predicted models by FARFAR2 (iii) and BRiQ refinement (iv) in the bottom panel.

models[24] but only for those 12 RNA puzzles with RMSD of best FARFAR2-predicted models less than 6 Å as shown in Fig. 4C. This is because it is not possible to refine those models that are far from native. For each puzzle, we select 2000 lowest energy models according to FARFAR2 (1547 models for PZ20) to run BRiQ refinement, and compare the best RMSD values of top 1% and 5% predicted models by FARFAR2 and BRiQ energy scores, respectively. As shown in Supplementary Data 4 and Fig. 4C, D, almost all (10/12) have reduced RMSD values after refinement with a median RMSD reduction of 0.48 Å for top 1% prediction and 0.45 Å for top 5% prediction. The only two RNA puzzles with increased RMSD values after refinement have very small changes (<0.08 Å for top 1% and <0.19 Å for top 5% predictions). When structural difference measured by DI (Supplementary Data 4), DI after BRIQ refinement were improved or maintained at the similar level for all 12 cases investigated. These results further indicate the consistency of BRiQ refinement.

As illustrative examples, Fig. 5 compares the conformations sampled by FARFAR2 and those further refined by BRiQ for Puzzles 4 and 18, respectively. Both show significant moves toward native structures of RNAs by comparing the dependence of energy scores on RMSD shown in Panels i and ii. Similar to the motif cases, BRiQ refinement yields improved backbone conformations over FARFAR2 models as shown in structural comparisons in Panels iii and iv of the figures.

## Discussion

This work presents a protocol for refining RNA model structures produced by other modeling tools. The protocol is nucleobase-centric in its knowledge-based energy score as well as the sampling algorithm. The fully knowledge-based score (BRiQ) captures the orientation dependence of base–base and base–oxygen, and oxygen–oxygen interactions by using a local coordinate system for each nucleobase, whereas the structures of ribose and

phosphate backbones were governed by rotameric statistical potentials and empirical internal energies. Moreover, the dominant base–base interaction was scaled by quantum mechanical calculations for removing possible indirect interactions contained in structural statistics. This statistical potential was coupled with the NuTree algorithm that samples the conformations around pre-determined base pairs. This refinement protocol improves 81% Rosetta-SWM models with less than 2 Å RMSD, 100% RNA puzzle models with RMSD < 4 Å, and 83% FARFAR2 models with RMSD < 6 Å. The CPU time cost of refining a single RNA puzzle model is around 40 min for a model of 50 nt, and 2 h for a model of 100 nt on Intel Xeon Gold 6242 CPU 2.80 GHz, and the RAM requirement is about 2.9 G.

The BRiQ statistical energy score is built on the knowledge that backbone conformations are determined by more stable stacked base pairs. This was done by focusing on base–base, base–oxygen, and oxygen–oxygen interactions while treating the backbone as rotameric states around the bases. The robustness of the scoring function is illustrated by the consistent improvement in refining a diverse set of RNA models produced from different methods participated in RNA puzzles including ModeRNA[6], FARNA[7], MC-Fold/MC-Sym[8], ifoldRNA[10,11], Vfold[12,13], 3dRNA[14,15], and SimRNA[16].

In order to secure as many RNA structures as possible, we included all RNA structures with resolution higher than 3.0 Å to collect base and backbone statistics for our BRiQ knowledge-based score. Although X-ray refinement at resolutions between 2.5 and 3.0 Å may contain errors and cryo EM structures may have poor density fitting, using 2.5 Å cut off and a limitation to X-ray structures will lead to 1225 structures with only 10.5% bases, a dataset that is too small to capture useful all-atom statistics. It is also noted that even for high-resolution structures, there are regions which have poor R-factor values or high clash scores. Structural regions with clashed atoms were automatically excluded from our statistics. Structural regions with poor R-factors simply reflect the regions that are more dynamic and the conformations of the regions are probable conformations among many. Thus, it is reasonable to incorporate these conformations as a part of statistics as commonly done for statistical potentials of proteins[26].

Another issue of the structural database is that the RNA structures included were dominated by rRNA and tRNA with about 89% base pairs from rRNAs. This raises the question whether or not our BRiQ energy function is biased toward rRNA and tRNA so that it would not be generalizable to other RNAs. In fact, RNA puzzles all are made of noncoding RNAs that are not involved in protein synthesis. Despite of this, BRiQ can consistently refine these noncoding RNAs, indicating that BRiQ is not an energy function limited to rRNA or tRNA.

The above demonstrated transferability is achieved because the statistics was made at base and atom levels, not at the structural motif level. Moreover, we are only interested in detailed energy surfaces around the local minima. That is, subtle or large conformational changes due to different ligands and crystallization conditions are useful to increase the resolution of the energy surface. In Supplementary Fig. S3, we plot the probability as a function of the pair orientation distance contributed by the structures of same sequences and by other structures. It is clear that the structures of same sequences can fill the conformational space missed by other structures, generating a more refined energy surface. To understand the potential impact of homologous structures to the refinement results of RNA puzzles, we removed all homologous structures of RNA puzzles with 70% sequence identity (the sequence similarity for the RNA "twilight zone" is 80%)[38]. We found that the homologous structures from RNA puzzles contributed only to 0.1% of all structural data. The

changes to the BRiQ energy score are negligible. Refinement results with the new BRiQ score are essentially the same (except those caused by stochastic nature of Monte Carlo sampling). For example, the difference before and after removing homologs for the refined FARFAR2 structures is only 0.02 Å for rp04 and 0.2 Å for rp18 for the best in top 1%, respectively. We will update the BRiQ energy score when more non-redundant RNA structures become available.

The NuTree algorithm samples the RNA conformations around predetermined base pairs from a given model. These preformed base pairs serve as the nucleus to speed up the folding of the rest of an RNA chain. However, if a base pair was incorrectly modeled, it will be energetically difficult to correct the mistake. This is the main reason why the current refinement protocol works best for those near-native structures, in which the majority of base pairs were correctly identified. Unlike Rosetta, the NuTree algorithm can fix both nested and non-nested (pseudoknot) base pairs. In other words, this algorithm will become more useful as secondary and tertiary base pairs are increasingly more accurately predicted by employing co-variational analysis of RNA homologous sequences[39–42] and deep learning techniques[43,44].

This work is limited to RNA structural refinement. Another question is whether or not the RBiQ statistical energy function coupling with the NuTree algorithm can serve as an effective method for ab initio structure prediction. Initial studies suggest that a new sampling algorithm is likely required because ab initio folding requires more frequent breaking and forming of base pairs than what are typically involved in the NuTree algorithm. The work in this area is in progress.

## Methods

### BRiQ energy score

*Base–base interaction energy* ($E_{bb}$). **Relative orientation between two bases**. The planar shaped bases make the orientation dependence a must for base–base interactions. Here, we consider each base $i$ as a rigid body with a local coordinate system $CS_i$ whose origin is located at atom C1'. The $x$-axis is in the direction of C1'N9 and the $z$-axis is in the direction of C1'N9×N9C4 for A and G whereas the $x$-axis is in the direction of C1'N1 and the $z$-axis is in the direction of C1'N1×N1C2 for U and C. As shown in Fig. 1A, the orientation and distance between bases $i$ and $j$ can be described by (1) the distance $r_{ij}$ between the origins of $CS_i$ and $CS_j$, (2) the rotational angle $\omega_{ij}$ around $r_{ij}$, (3) the directional vector $r_{ij}$ in $CS_i$, and (4) the directional vector $r_{ji}$ in $CS_j$ (a total of 6 dimensions). The distance $r_{ij}$ has a range of 0–15 Å with a uniform grid space of 0.3 Å. The rotational angle $\omega_{ij}$ varied from −180° to 180° with a uniform grid of 8°. We represent the orientation of the directional vector $r_{ji}$ by 2000 uniformly distributed points on a sphere from Monte Carlo simulated annealing of points with repulsive interactions proportional to the inverse of squared distances between the points. Thus, the space between the two bases is separated into $50 \times 45 \times 2000 \times 2000$ discrete regions. Once the energy values for all these regions are known, the energy value at a given distance and orientation can be linearly interpolated. The above coordinate system for representing the base–base relative orientation was similar to but is more sophisticated than what was proposed for a coarse-grained two-particle representation of RNA chain[45]. The 6-dimensional base–base interactions were pre-calculated and stored in a table and only negative values were loaded into computer memory for quick access.

**Base-orientation representation**. To define an orientation-dependent density, it is necessary to measure the structural similarity first. One common measure is the root-mean-squared distance (RMSD) between two base pairs. However, it is too time-consuming to calculate RMSD. To speed up the calculation, we assumed that four fixed points (T1, T2, T3, T4) in the local coordinate system of a base (Supplementary Fig. S4a) are sufficient to represent the orientation of each base and the mean squared distance between these points in two bases A and B (DDM) can approximate RMSD with DDM defined by

$$DDM = \sqrt{\sum_{i=1}^{4}\sum_{j=1}^{4} d(T_{Ai}, T_{Bj})^2 / 16} \qquad (2)$$

We have optimized T1, T2, T3, T4 so that DDM has the highest correlation coefficient to RMSD. The final four points in each base coordinate system are: T1 = (2.158, 3.826, 1.427), T2 = (−0.789, −0.329, −1.273), T3 = (4.520, −3.006, 1.586) and T4 = (6.018, 1.903, −1.638). The resulting correlation coefficient is 0.974 (Supplementary Fig. S4b).

**Orientation distribution density**. We employed a modified radial basis function kernel $h(d)$ to calculate the orientation distribution of one base around

another base $f(x)$ by the following equations:

$$h(d) = \begin{cases} 1, \text{if } d \leq 0.15 \\ e^{-0.5((d-0.15)/0.1)^2}, \text{if } d > 0.15 \end{cases} \quad (3)$$

and

$$f(x) = \sum_{i=1}^{N} h(DDM(x, x_i)) \quad (4)$$

where $N$ is the number of base pairs in the database of RNA structures and DDM $(x, x_i)$ is the distance between a target base pair and a base pair in the database. If $DDM(x, x_i)$ between two bases is less than 0.15 Å, we consider two bases shared the same orientation. Here and below, we have employed an empirical value (0.1) for defining the spread of the kernel based on statistics from the PDB structures and a few trials. An example is shown in Supplementary Fig. S5.

The orientation distribution densities of base pairs with different sequence-separation distances (1, 2 or >2) were calculated separately. For non-local base pairs (separation >2), the density was re-weighted by quantum mechanical energy.

**Quantum-mechanical-energy-weighted orientation distribution density.** One issue associated with a statistical energy function is that a high population of specific orientation between two bases may not be due to a strong direct interaction at this specific orientation but due to the orientation preference of the interaction with another base. Here, we minimize the possibility of this type of indirect interactions by a weighting factor $w(x_i)$ according to quantum calculations. The assumption is that the strength of the directional interaction is related to the strength of the quantum interaction energy. Here,

$$w(x_i) = e^{-\frac{E_{QM}(\tilde{x}_i)}{4.32\text{kcal/mol}}}/f(\tilde{x}_i) \quad (5)$$

where $\tilde{x}_i$ is the nearest representative configuration to $x_i$, $E_{QM}(\tilde{x}_i)$ is the quantum energy. The modified orientation distribution $f'(x)$ satisfies

$$f'(x) = \sum_{i=1}^{N} w(x_i) h(DDM(x, x_i)) \quad (6)$$

Here, quantum mechanical (QM) calculations were performed by using Guassian 09[46]. We investigated all possible ten pairs of bases (AA, AU, AG, AC, UU, UG, UC, GG, GC, and CC). That is, both canonical (AU, GC, GU) and all other possible noncanonical pairs were included in the statistics. Moreover, for each base-pair type, we generated 80 orientation cluster centers by minimizing the root mean square distance between all data points to the nearest cluster center. That is, not only base pairs, but base–base stacking and other base–base polar interactions were included in the statistics. For each representative configuration in a structural cluster, we employed 5 nearest base-pairing conformations with the highest-resolution PDB structures as the initial configurations for QM calculations. The initial structure from the PDB was further optimized quantum mechanically so as to minimize the effect of potentially inaccurate conformations. The steps for calculating QM base–base interactions are as follows: (1) remove all backbone atoms, (2) replace the C1' atom with a H atom, (3) optimize the position of the H atom by ab initio Hartree–Fock calculations with the basis set 6-31 G* and (4) calculate the base–base interaction energy by $E(AB) = H(AB) - H(A) - H(B)$ with the density-functional theory method M06-2X[47] and the basis set 6-31+G(d,p). The average value of five QM calculations is considered as the QM energy for the representative configuration. A total of 4000 QM calculations for 10 possible base pairs ($80 \times 10 \times 5$) were performed. Figure 1C shows some examples of the calculation results. The scaling parameter (4.32 kcal/mol) in Eq. (5) between QM calculations and statistical energy functions is obtained from the slope of the regression analysis of hydrogen-bonded base pairs (Fig. 1C).

**Base–base interaction energy.** The final energy function for base–base interactions is calculated by the following equations:

$$E_0(x, \text{sep}) = -\ln f'(x)/f_{\text{ref}}(\text{sep}) \quad (7)$$

and,

$$E_{bb}(x, \text{sep}) = \begin{cases} E_0(x, \text{sep}), \text{if } E_0(x, \text{sep}) < 0 \\ 0, \text{if } E_0(x, \text{sep}) \geq 0. \end{cases} \quad (8)$$

where sep is the sequence separation distance, its value could be 1, 2 or 2+. $f_{ref}(2+)$ was employed to scale the lowest energy value of non-local base pair to −8.0 based on the lowest energy from QM calculations and the scaling between statistical and QM calculations (Fig. 1C). $f_{ref}(1)$ and $f_{ref}(2)$ were employed to scale the lowest energy value of local base pair to −4.0. Here, we also set the maximum base–base interaction energy to zero because repulsive interactions from the above six-dimensional statistical analysis are not reliable. Figure 1B shows a schematic example of before and after QM scaling.

*Interaction energy between a base and a main-chain oxygen atom* $E_{bo}(x)$. The interaction energy between a base and a main-chain oxygen atom is the interaction between a polar group and a polar atom. The orientation of a base is still represented by 4 points obtained previously. The relative orientation and position between an oxygen atom and a base can be described by the distance between the oxygen atom and the four points representing the base. The kernel function h(d)

and the density f(x) for $E_{bo}(x)$ are given by

$$h(d) = e^{-0.5(d/0.16)^2} \quad (9)$$

and

$$f(x) = \sum_{i=1}^{N} h(d(x, x_i)) \quad (10)$$

where $x$ and $x_i$ represent the target base–oxygen and a base–oxygen pair in the database, respectively, and $d(x, x_i)$ are the RMSD between the two base–oxygen pairs. Unlike $E_{bb}$, the repulsive (positive) interaction was not set to zero but reduced by a decaying weighting function $w_{bo}(d_{bo}^{\min}) = 1 - 1/(1 + e^{(3.7 - d_{bo}^{\min})/0.08})$, where $d_{bo}^{\min}$ is the minimum distance between an oxygen atom and any atoms in a base, 3.7 Å is the approximate interaction distance between a mainchain oxygen atom and a base and 0.08 is an empirical parameter to control the rate of decay. We also impose an orientation dependence for the hydrogen bonding between an oxygen atom and the corresponding atoms in a base according to the $\theta$ angle between the base atom N or O, the main chain atom O (O2', OP1, OP2) and the main chain atom C or P bonding to O. This was done by locating the angle range in RNA structures after removing top 3% angles from each side and defining $s(\theta) = 0$ for outside the angle range, $s(\theta) = 1$ when $\theta = \theta_M$, the median value of $\theta$, and

$$s(\theta) = \begin{cases} 1 - \left(\frac{\theta - \theta_M}{\theta_L - \theta_M}\right)^2, \text{if } \theta_L < \theta < \theta_M \\ 1 - \left(\frac{\theta - \theta_M}{\theta_U - \theta_M}\right)^2, \text{if } \theta_M < \theta < \theta_U \end{cases} \quad (11)$$

where $\theta_U$ and $\theta_L$ are the upper and lower bounds for $\theta$, respectively. The final expression for $E_{bo}(x)$ is

$$E_{bo}(x) = -w_{bo}(d_{bo}^{\min})s(\theta)\ln f(x)/f_{\text{ref}} \quad (12)$$

where $f_{\text{ref}}$ is employed to scale the lowest energy value empirically to −3.0, which we set to be about one-third strength of GC pairs. As an example, Fig. 1D shows the distribution of a nucleobase C and an oxygen atom in a phosphate group and that of a nucleobase A and O4 in ribose.

*Hydrogen-bonding energy between main-chain oxygen atoms* $E_{oo}(x)$. We consider hydrogen bonding interactions between O2'-O2', O2'-OP and OP-OP. The configuration of hydrogen bonds is determined by four atoms (two oxygen atoms and their connecting C or P atoms). The distance $d$ between two hydrogen-bonding configurations is calculated by RMSD between the four atoms. Similar to $E_{bo}(x)$, we have

$$E_{oo}(x) = -w_{oo}(d_{oo})\ln\frac{f(x)}{f_{\text{ref}}} \quad (13)$$

with $f(x) = \sum_{i=1}^{N} e^{-0.5(d(hb, hb_i/0.1)^2}$, $f_{\text{ref}}$ is employed to scale the lowest energy value empirically to -3.0 for O2'-O2', -2.0 for O2'-OP, -1.5 for OP-OP and the decay function $w_{oo}(d_{oo}) = 1 - 1/(1 + e^{(3.3 - d)/0.07})$ with 3.3 Å is the typical hydrogen bond length between two oxygen atoms and 0.07 is the empirical parameter to control the decay rate. Here the angle dependence is implicitly accounted for by using 4 atomic coordinates (e.g., atoms C2'-O2'-OP-P for the hydrogen bond between O2' and OP) to define RMSD.

*Energy for atomic clashes* ($E_{\text{clash}}$). Due to lack of adequate statistics for hard-core exclusion in all statistical energy functions, we have set repulsive terms to 0 in $E_{bb}(x)$ or quickly decay to 0 by weighting functions $w_{bo}(d_{bo}^{\min})$ and $w_{oo}(d_{oo})$ in $E_{bo}(x)$ and $E_{oo}(x)$, respectively, when the distance is less than a preset value. To avoid direct atomic clashes, we introduce an empirical energy function as below.

$$E_{\text{clash}}(d) = \begin{cases} (k_{\text{clash}}*0.4)^4 - 4(k_{\text{clash}}*0.4)^3(d - r_0 + 0.4), \text{if } d \leq r_0 - 0.4, \\ (k_{\text{clash}}*0.4)^4, \text{if } r_0 - 0.4 < d < r_0, \\ 0, \text{if } d > r_0, \end{cases} \quad (14)$$

where $d$ is the atomic distance between two atoms, $r_0$ is the shortest statistical distance from RNA structures between the two atoms and $k_{\text{clash}}$ is an empirical parameter to control the increasing rate of repulsion when two atoms approach to each other. We set $k_{\text{clash}} = 3$ after the energy minimized structure does not change much for $k_{\text{clash}}$ between 2 and 5. $r_0$ is orientation independent for atoms in SP3 hybridization but is orientation-dependent for atoms in SP2 hybridization. That is, $r_0$ is orientation-dependent between any atom with an atom in a base but is orientation-independent between two main-chain atoms. For orientation-independent $r_0$, it is set as the top 5% shortest distance between two atoms found in RNA structures. For orientation-dependent $r_0$ between a given atom a and an atom b in base B, the atomic position of $a$ is transformed to the local coordinate system of the base B and clustered around the atom b. The top 5% shortest distance in different orientations is set as $r_0$ in different orientations. The functional form of the clash energy is shown in Supplementary Fig. S6.

*Main-chain rotameric energy* ($E_{rot}$). For protein structure prediction, the sidechain conformational space is discretized into rotamers. Here, we introduce main-chain rotameric states because stacked bases are more rigid than the main chains. The

conformational space of a ribose rotamer is mainly determined by two dihedral angles (Fig. 1E). The first is the rotational angle $\chi$ rotating around the chemical bond connecting the base and ribose (N9-C1' for bases A and G and N1-C1' for bases U and C). The second is the improper angle $\nu$ between the C2'-C4'-O4' plane and the C2'-C4'-C3' plane within the connected ribose. Each dihedral angle can be separated into two regions, and each region into 300 to 500 rotamers, and a total of 1500 rotamers is employed to describe a ribose rotameric state.

The ribose rotamers are clustered according to RMSD. This is done by transforming atomic coordinates of eight ribose atoms to the local coordinate system for the base. RMSD is the average distance of ribose atoms in the local coordinate system. Once RMSD is known, we can use the kernel density estimation method to calculate the density of each rotamer, and then take the negative logarithm and convert it to the rotamer energy $E_{rot}(x)$ as below.

$$E_{rot}(x) = -\ln\frac{f(x)}{f_{max}} \quad (15)$$

where $(x) = \sum_{i=1}^{N} e^{-0.5(d(x,x_i)/0.15)^2}$, $x_i$ denotes a rotamer in the database and the summation is over all the rotamers in the database.

*The internal energy $E_{internal}$.* We introduce the following internal energies ($E_{internal}$) between a phosphate group and two connecting riboses to account for energetics associated to changes in bond lengths $E_{bond}$, bond angles $E_{angle}$, and dihedral angles $E_{torsion}$

$$E_{bond}(u) = \begin{cases} -2u-1, & if\, u < -1, \\ u*u, & if -1 \leq u \leq 1, \\ 2u-1, & if\, u > 1 \end{cases} \quad (16)$$

where $u(bl) = k_{bond}(bl - 1.422)$ and $bl$ denotes the bond length between O5'$_{i+1}$ and C5'$_{i+1}$. We made this bond slightly flexible so that we can add a phosphate group (atom P, OP1, OP2, O5') between two fixed riboses at fixed dihedral angles, other bond lengths and bond angles. The average bond length found in the RNA structures (1.422 Å) is employed here. Parameter $k_{bond}$ is set to 5.0, which is optimized by backbone modeling (rebuilt main-chain atoms after fixing base positions).

$$E_{angle}(u) = \begin{cases} -2u-1, & if\, u < -1, \\ u*u, & if -1 \leq u \leq 1, \\ 2u-1, & if\, u > 1 \end{cases} \quad (17)$$

where $u(\theta_{POC}) = k_{angle}(\theta_{POC} - 120.7)$ and $u(\theta_{OCC}) = k_{angle}(\theta_{OCC} - 111.1)$. $\theta_{POC}$ is between $_{Pi+1}$-O5'$_{i+1}$-C5'$_{i+1}$ and $\theta_{OCC}$ is between O5'$_{i+1}$-C5'$_{i+1}$-C4'$_{i+1}$. Similar to the bond length, we made these two angles flexible so that we can add a phosphate group between two fixed riboses at fixed dihedral angles, other bond lengths and bond angles. The average values for these two angles in RNA structures are 120.7° and 111.1°, respectively. Parameter $k_{angle}$ is set to 0.1, after a few trials of backbone modeling. For both bond-angle and bond-length energies, a harmonic function with linear extrapolations at both ends was employed to improve conformational sampling efficiency.

$E_{torsion}$ is calculated from three statistical energy functions (see Fig. 1E for angle definitions).

$$E_{torsion} = E(\nu_i, \varepsilon_i, \varsigma_i) + E(\varsigma_i, \alpha_{i+1}, \beta_{i+1}) + E(\beta_{i+1}, \gamma_{i+1}, \nu_{i+1}) \quad (18)$$

with $E(\nu_i, \varepsilon_i, \varsigma_i) = -\ln P(\varepsilon_i, \varsigma_i | \nu_i)$, $E(\varsigma_i, \alpha_{i+1}, \beta_{i+1}) = -\ln P(\alpha_{i+1} | \varsigma_i, \beta_{i+1})$, and $E(\beta_{i+1}, \gamma_{i+1}, \nu_{i+1}) = -\ln P(\beta_{i+1}, \gamma_{i+1} | \nu_{i+1})$. Here, two improper angles ($\nu_i$ and $\nu_{i+1}$) belong to the two neighboring riboses connected by a phosphate group. Two dihedral (angles $\varepsilon_i$ involving C2'$_i$-C3'$_i$-O3'$_i$-$P_{i+1}$ and $\varsigma_i$ involving C3'$_i$-O3'$_i$-$P_{i+1}$-O5'$_{i+1}$) determines the position of the phosphate group. In addition, $\alpha_{i+1}$, $\beta_{i+1}$, and $\gamma_{i+1}$ dihedral angles involve O3'$_{i-1}$-$P_{i+1}$-O5'$_{i+1}$-C5'$_{i+1}$, $P_{i+1}$-O5'$_{i+1}$-C5'$_{i+1}$-C4'$_{i+1}$ and O5'$_{i+1}$-C5'$_{i+1}$-C4'$_{i+1}$-O4'$_{i+1}$, respectively[48]. Here, the coupling between neighboring three dihedral angles, rather than the statistics of a single torsion angle was considered to improve the accuracy of main-chain modeling.

*Structure database.* All statistical energy terms in Eq. (1) were derived from 2247 RNA structures obtained by X-ray crystallography and cryogenic electron microscopy with a resolution higher than 3.0 Å downloaded on January 23, 2020 from the PDB databank[1]. This set contains 272 ribosome, 221 riboswitches, 138 tRNA, 106 ribozymes, 42 aptamers, 121 virus RNA, 17 introns, 3 spliceosomes, and 1327 others. Among them, there are 1459 protein–RNA complex structures and 788 RNA-only structures. The base pairing information is dominated by ribosome (about 89%). We did not use the Cambridge database because it contains simple structures only. We did not remove any redundant sequences because we need a database as large as possible. Moreover, RNAs can have different structures in different complexes and we want to capture all possible conformations. We can do this because statistics are collected at the atomic or base level. See the discussion in more details.

## Conformational sampling algorithm
*Confirmational representation by the NuTree.* For conformational sampling, each RNA structure is represented by the NuTree. Each node in the NuTree denotes an

RNA residue including its base position (the local coordinate system), the ribose rotameric state and the phosphate position attached to the 3' position of the ribose (determined by dihedral angles $\varepsilon$ and $\zeta$, Fig. 1E). The edge of the NuTree represents relative positions between the local coordinate systems of two bases, which are described by a $3 \times 4$ coordinate transformation matrix. There are ten types of coordinate transformation matrices to describe nine types of edges: (1) sequence-neighboring base pairs along 5' to 3' and 3' to 5' directions in the loop regions, (2) sequence-neighboring base pairs along 5' to 3' and 3' to 5' directions in the Watson–Crick pairing regions, (3) sequence-neighboring base pairs along 5' to 3' and 3' to 5' directions in the non-Watson–Crick pairing regions or connection between Watson–Crick pairing and non-Watson–Crick pairing regions, (4) Watson–Crick pairs, (5) Non-Watson–Crick pairs, (6) chain connection without a specific positional relation (jump). Except for jump, each edge has a corresponding move set. The move set was collected from all possible conformations of the same type in the structural database that were discretized into 60 representative conformations and each representative conformation was further discretized into 60 sub-representative conformations. These 3600 representative conformations form the move set for each edge type. Figure 1F shows the NuTree for the GCAA tetraloop with a sequence of $G_0C_1G_2C_3A_4A_5G_6C_7$ where the edge 0–1 is sequence-neighboring base pairs along the 5' to 3' direction in the Watson–Crick pairing region, edges 1–2 is sequence-neighboring base pairs along the 5' to 3' direction in non-Watson–Crick pairing regions, 2–3, and 3–4 are sequence-neighboring base pairs along the 5' to 3' direction in the loop regions, edges 0–7, 1–6 are Watson–Crick pairs and 2–5 is non-Watson–Crick pair. Although there is no edge between 4 and 5, we need to build the phosphate connection and calculate the internal energy between them.

For constructing a RNA model, the average coordinates were used for atoms in the bases. Atoms in riboses were generated from representative PDB conformations. Bond length and bond angles associated to the phosphate group were set to the average value in PDB library. It should be emphasized that atomic coordinates of riboses were not generated from a fixed bond length and angles but directly from the pucker-dependent conformers[49] from PDB structures.

*Conformational sampling moves.* Conformational sampling involves node and edge sampling. Node sampling allows either a random selection of ribose rotamers or a small adjustment of the base local coordinate system (<0.2 Å translational and <2° rotational motions). Both moves are local. They do not change the positions of other nodes and riboses. Edge sampling makes both local and global moves. The local move makes minor adjustment (<0.2 Å translational and <2° rotational motions). The global move randomly selects a move from the set according to the edge type. Edge sampling moves will change the positions of all downstream nodes but not the relative positions between the nodes.

The above sampling may change the relative positions of some neighboring riboses and make it necessary to rebuild the phosphate connection. Such connection can be rapidly built by the grid search from representative points for the lowest $E_{internal}$ value in the $\varepsilon$–$\zeta$ torsional angle space. According to the distribution of conformations (Supplementary Fig. S7), we have divided $\varepsilon$–$\zeta$ space into 50 representative points, each representative point into 50 sub-representative points and each sub-representative point with 25 local points in 1° extension. That is, a total of 125 calculations is conducted for searching the lowest $E_{internal}$ value.

*Monte Carlo simulated annealing.* All nodes in a NuTree are divided into three types: those with positions fixed (A), those with relative internal positions fixed but not absolute positions (B), and those with both relative and absolute positions changed (C). The energy change after each move is calculated by

$$\triangle E = dE(AB) + dE(AC) + dE(BC) + dE(CC) + dE_{rot} + dE_{internal} \quad (19)$$

where $dE(AB)$, $dE(AC)$, $dE(BC)$, and $dE(CC)$ denote the changes in interaction energies with the AB, AC, BC, and CC regions, respectively, and $dE_{rot}$ and $dE_{internal}$ are the changes in rotameric and internal energies, respectively. Each move is accepted or rejected according to the Metropolis criterion. We employed simulated annealing for energy optimization with an initial temperature set at 2.5, which is decreased by a factor of 0.95 at each round until temperature reached 0.01. In the refinement protocol, the initial temperature is set at 0.5 which is decreased to 0.01 by a factor of 0.9. The sampling steps at each temperature $N_{step}$ is proportional to the edge number in the NuTree.

$$N_{step} = 400*n(\text{wc edge}) + 2000*(nwc\ edge) + 4000*n(\text{other edge}) \quad (20)$$

where $n$(wc edge) is the edge number of Watson–Crick pair or helix neighbor, $n$ (nwc edge) is the edge number of Non-Watson–Crick pair or NWC neighbor, $n$ (other edge) is the number of other edges.

To increase the efficiency of conformational sampling, we give $E_{clash}$ and $E_{internal}$ a low weight of 0.05 at the initial temperature and gradually increase the weight to 1.

*Motif test set.* The test set is obtained from the benchmark of Rosetta-SWM[34], downloaded from https://purl.stanford.edu/fq893cm4516. After removing redundant RNAs and those with more than 50 bases and non-standard RNA bases, we have 48 motifs listed in Supplementary Data 1.

*RNA puzzle and FARFAR2 test sets.* RNA Puzzle data set is downloaded from https://github.com/RNA-Puzzles/raw-dataset-and-for-assessment. The prediction results of FARFAR2[24] are downloaded from https://purl.stanford.edu/wn364wz7925.

**Reporting summary**. Further information on research design is available in the Nature Research Reporting Summary linked to this article.

## Data availability
The structural data and test sets used by BRiQ is publicly available at http://servers.sparks-lab.org/downloads/BRiQ-dataset.tar.gz.

## Code availability
The source code of BRiQ refinement is available at https://github.com/Jian-Zhan/RNA-BRiQ. We have also made the code citable by obtaining a DOI for the Github repository, which allows a permanent reference to the version of the code used in this study[50].

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

## Acknowledgements
This work was supported by Australian Research Council DP180102060 and DP210101875 to Y.Z. We also gratefully acknowledge the use of the High Performance Computing Cluster 'Gowonda' to complete this research, and the aid of the research cloud resources provided by the Queensland CyberInfrastructure Foundation (QCIF). We gratefully acknowledge the support of NVIDIA Corporation with the donation of the

Titan V GPU used for this research. P.X. would like to thank the hospitability of Shenzhen Bay Laboratory while he is a visiting scholar to the laboratory. The support of Shenzhen Science and Technology Program (Grant No. KQTD20170330155106581) and the Major Program of Shenzhen Bay Laboratory S201101001 is acknowledged.

## Author contributions

P.X. designed and performed the method and wrote the manuscript, R.W. assisted in quantum-mechanical calculations, J.Z. and Y.Z. conceived of the study, participated in the initial design, and assisted in data analysis. Y.Z. drafted the whole manuscript. All authors read, contributed to the discussion, and approved the final manuscript.

## Competing interests

The authors declare no competing interests.
