## [Peer Review File · Nature Communications]

Reviewers' Comments:

Reviewer #1:

Remarks to the Author:

The manuscript describes BRiQ, a new method for refining the RNA 3D structure models that can be used to improve the 3D structures predicted by various algorithms. BRiQ is based on a knowledge-based potential that has been developed specifically for RNA, focusing on base pairing and stacking interactions. The method features an additional algorithm for sampling backbone conformations in the neighborhood of RNA basepairs. Using BRiQ it is possible to improve many of the 3D models submitted to the RNA Puzzles project. As RNA 3D structure prediction and model refinement are far from being a solved problem, any improvement in this field is very welcome. It is particularly encouraging that BRiQ is geared specifically towards RNA 3D structure instead of adapting protein-specific approaches. The manuscript is well-written and is easy to follow; however, the paper raises several questions listed below.

Major comments

1) Non-canonical basepairs are critical for RNA 3D structure and function, and the RNA Puzzles project identified the need to improve the prediction and modelling of the regions structured by the non-Watson-Crick basepairs. Although the manuscript does mention the non-canonical basepairs, it does so only in passing and it is not clear how they are factored into the method.

For example, in the section "Quantum-mechanical-energy-weighted orientation distribution density", the authors describe how the 4,000 QM calculations were made. For each of the 10 possible pairs, there were 80 representative configurations and the average value of five QM calculations is taken into account ($10 \times 80 \times 5$). However, it is not clear why only 10 possible pairs are being analysed? The seminal work by Leontis and Westhof (PMID:11345429) identified 12 families of RNA basepairs. It appears that in this model several geometric basepairs families are grouped together, which could have a negative effect on the refinement accuracy for non-canonical basepairs.

The manuscript would benefit from a discussion of how well BRiQ performs with non-canonical basepairs, and what measures are taken to refine the regions that are structured by non-canonical basepairs, such as hairpins, internal loops, and junction loops.

2) Did the authors consider applying the metric proposed by Parisien et al in addition to RMSD (PMID:19710185)? The use of the Deformation Index would help evaluate how well the method performs with respect to improving base-base interactions, especially since this is supposed to be the BRiQ's key strength.

3) It remains unclear why redundancy reduction is not necessary. The authors suggest that "a few redundancies will have an insignificant effect on all statistics collected at the atom or base level", and they test this assumption by clustering the sequences used in RNA Puzzles or FARFAR2 structures. However, I am not convinced that this is a good test because if the authors include all high-resolution RNA structures to collect base and backbone statistics, then the calculations are heavily biased towards the rRNA and tRNA structures representing the vast majority of 3D structures. Also many rRNA 3D structures are heavily redundant as they use the same template 3D structure and only differ in a region of interest (for example, an antibiotic binding site). In addition, high resolution is not the only measure of quality, as it could be important to consider that local regions of high quality 3D structures could have poor R-factor values or high clash scores.

Minor comments

Did the authors consider including in Figure 3 an internal loop that was improved by the method, such as loop E or a kink-turn? While having two widely occurring, albeit simple, hairpins is useful, an

internal loop example will also be helpful for demonstrating method's performance.

I wonder if the PDB structures that were part of the RNA Puzzles project have been excluded from the set used to calculate the BRiQ's parameters? If not, then it's conceivable that the improvement in the RNA Puzzles examples was due to the fact that the target structures have been seen by the method. As RNA Puzzles participants are likely to be the future users of BRiQ, it is important to show that BRiQ does well even for previously unknown 3D structures.

I downloaded the source code and tried installing it on my Mac, however, running "cmake ../" resulted in the following error:

```
BRiQ/build/CMakeFiles/FindOpenMP/OpenMPTryFlag.cpp:2:10: fatal error: 'omp.h' file not found
#include <omp.h>
^~~~~~
1 error generated
```

It could be useful to document all the requirements (for example, the installation instructions refer to cmake but it's not listed as a requirement).

The user documentation found in the Readme file is very limited and should be expanded (currently it contains just 55 lines, including blank lines).

The version of the software should be specified in the manuscript, and different versions of the software should be hosted separately to ensure the reproducibility of the work described in the paper. Did the authors consider hosting the source code on GitHub, BitBucket or similar platform to facilitate version tracking?

Typos

Should "Inter Xeon Gold" be "Intel Xeon Gold"? Also, in addition to the CPU used for performance testing it will be useful to state the RAM requirements.

Reference 43 does not appear to be formatted correctly.

Reviewer #2:
See Attached

The article « Robust RNA structure refinement by a nucleobase-centric sampling algorithm coupled with a backbone rotameric and quantum-mechanical-energy-scaled base-base knowledge-based potential » by Peng Xiong, Ruibo Wu, Jian Zhan and Yaoqi Zho describes a refinement protocol for modelled RNA structures. The topic is important, and the paper is well written, clear and logically organized. The improvements are real, although not overwhelming.

My points are the following.

1/ The authors derived the statistical energy terms from “2247 RNA structures obtained by X-ray crystallography and cryogenic electron microscopy with a resolution higher than 3.0Å downloaded on January 23, 2020”. I guess this was from the PDB, a reference is needed. Now the cut-off at 3.0 Å is too high. X-ray refinement at resolutions between 2.5 and 3.0 Å are not easy and contain many errors. There is no point of making highly sophisticated approximations and quantum-mechanical calculations on rather approximate structures with in-built errors. In this respect, the use of cryo EM structures is inappropriate since those used previous structures and are fitted in density and often with poor refinement (since only recently respectful refinement programs have been used). I realize they need lots of structures but adding noise (often in a redundant fashion) will not improve the parameters. They do state that they did not remove redundant sequences and, thus, several structures must be ribosomal structures, a lot of which are stereochemically full of errors because of the difficulties in data and the use of previous structures. There are only 1528 RNA structures present in the PDB now. So, I guess the authors used also RNP structures. What is the number of RNP structures? Are they all ribosomes? Do they include spliceosomes? In any case, one should only consider structures below 2.5 Å especially for feeding into quantum chemical programs. Do they use structures from the Cambridge data base? They are simpler structures but high resolution. The NDB contains also a lot of data on nucleic acids that are well curated.

2/ What would be the results in terms of RMSDs if you submit to re-refinement some of the structures between 2.5 and 3.0 Å? Or those below 2.0 Å resolution (hoping some are present)?

3/ Which dictionaries for bond lengths and angles were used? The dictionaries are not always standardized and thus, how variations in these fundamental geometrical parameters influence the final parameters and refinement?

4/ “the interactions associated with bases and oxygen atoms in backbones are strongly orientation-dependent.” Indeed true, but bond lengths and angles are also conformation dependent; this is clearly seen in ribose dimensions depending on the sugar pucker (see JACS 102, 1493 (1980)). Are these variations considered? Such a level of precision should be needed for quantum calculations.

5/ MolProbity is used regularly for checking RNA structures. Is there an improvement in the clash score after re-refinement? For example, in some RNA Puzzles the clash scores are high. Are there improved and if yes by how much? New Tables may be necessary (can be in sup mat).

6/ Minor. In the abstract “the progress in protein or RNA structure refinement has been slow because native structures are often not the global minimum of existing approximate energy scores.” I think it should be stated the other way around: the global minimum given by the energy scores is not at the experimentally determined “native” structure.

Reviewer #1 (Expertise: RNA structural prediction):

Major comments

1) Non-canonical basepairs are critical for RNA 3D structure and function, and the RNA Puzzles project identified the need to improve the prediction and modelling of the regions structured by the non-Watson-Crick basepairs. Although the manuscript does mention the non-canonical basepairs, it does so only in passing and it is not clear how they are factored into the method.

For example, in the section “Quantum-mechanical-energy-weighted orientation distribution density”, the authors describe how the 4,000 QM calculations were made. For each of the 10 possible pairs, there were 80 representative configurations and the average value of five QM calculations is taken into account ($10 \times 80 \times 5$). However, it is not clear why only 10 possible pairs are being analysed? The seminal work by Leontis and Westhof (PMID:11345429) identified 12 families of RNA basepairs. It appears that in this model several geometric basepairs families are grouped together, which could have a negative effect on the refinement accuracy for non-canonical basepairs.

The manuscript would benefit from a discussion of how well BRIQ performs with non-canonical basepairs, and what measures are taken to refine the regions that are structured by non-canonical basepairs, such as hairpins, internal loops, and junction loops.

Answer: We would like clarify that 10 base-pair types included all possible pairs (AA, AU, AG, AC, UU, UG, UC, GG, GC and CC). That is, both canonical (AU, GC, GU) and all other possible noncanonical pairs were included in the statistics. Moreover, for each base-pair type, we generated 80 orientation cluster centers by minimizing the root mean square distance between all data points to the nearest cluster center. That is, not only base pairs, but base-base stacking and other base-base polar interactions were included in the statistics. We now added additional clarifications in the method section.

Per suggestion, in addition to the whole motif, we further analyzed the refinement results of RNA puzzles at the base-pair level. Here, we employed DDM to measure the relative orientational difference between predicted and native base pairing structures according to four pseudo atoms employed for representing each base (see Methods). As the new Supplementary Table S3 shows, the average DDM values from

native base-pairing structures of Watson-Crick pairs, non-Watson-Crick pairs, and base-stacking decreased 30% from 0.545 to 0.384, 17% from 0.687 to 0.570, and 22% from 0.834 to 0.650, respectively. The improvement in base pairing structures after BRiQ refinement is found for essentially all RNA puzzles (except for non-Watson-Crick pairs in PZ02, PZ03, and PZ05), regardless whether or not there is an improvement of the overall RMSD or not. More improvement at the base-pair level indicates that the BRiQ refinement occurred at the detailed atomic resolution.

2) Did the authors consider applying the metric proposed by Parisien et al in addition to RMSD (PMID:19710185)? The use of the Deformation Index would help evaluate how well the method performs with respect to improving base-base interactions, especially since this is supposed to be the BRiQ's key strength.

Answer: Thank you for the suggestion. We now included deformation index (DI) results in Supplementary Tables S2 and S4 for the RNA puzzles and FARFAR2 test set, respectively. Indeed, we found that more RNAs showed the improvement in DI after refinement than in RMSD. For example, 20/25 predicted DI values in RNA puzzles were reduced, compared to only 13/25 in predicted RMSD. For FARFAR 2 dataset, DI after BRiQ refinement were improved or maintained at the similar level for all 12 cases investigated. We now reported it in Supplementary Tables S2 and S4 and described it in the results section.

3) It remains unclear why redundancy reduction is not necessary. The authors suggest that "a few redundancies will have an insignificant effect on all statistics collected at the atom or base level", and they test this assumption by clustering the sequences used in RNA Puzzles or FARFAR2 structures. However, I am not convinced that this is a good test because if the authors include all high-resolution RNA structures to collect base and backbone statistics, then the calculations are heavily biased towards the rRNA and tRNA structures representing the vast majority of 3D structures. Also many rRNA 3D structures are heavily redundant as they use the same template 3D structure and only differ in a region of interest (for example, an antibiotic binding site). In addition, high resolution is not the only measure of quality, as it could be important to consider that local regions of high quality 3D structures could have poor R-factor values or high clash scores.

Answer: Thank you for the question. We agree that in a perfect world we would use all non-redundant structures in super high resolutions for every atom. However, we have to live with what we can get (only a few nonredundant RNA structures solved) and made every attempt to avoid overfitting through the design of the energy score. The reviewer's main concern is whether or not our energy function is biased toward rRNA and tRNA so that it would not be generalizable to other RNAs. In fact, RNA puzzles all are made of noncoding RNAs that are not involved protein synthesis. Despite of this, BRiQ can consistently refine these noncoding RNAs, indicating that BRiQ is not an energy function limited to rRNA or tRNA. This transferability is achieved because the statistics was made at base and atom levels, not at the structural motif level. Moreover, we are only interested in detailed energy surfaces around the local minima. That is, subtle or large conformational changes due to different ligands and crystallization conditions of the same RNAs are useful to increase the resolution of the energy surface. In the new Supplementary Figure S3, we plot the probability as a function of pair orientation distance contributed by the structures of same sequences and by other structures. It is clear that the structures of same sequences can fill the conformational space missed by other structures, generating a more refined energy surface. Finally, we would like to point out that using a resolution cut off for selecting the structures is a common practice in generating statistical potentials from protein or RNA structures. Structural regions with clashed atoms were automatically excluded from our statistics. Structural regions with poor R-factors simply reflect that the regions are more dynamic and the conformations of the regions are probable conformations among many. Thus, it is reasonable to incorporate these conformations as a part of statistics. This is now discussed in the discussion section.

Minor comments

1) Did the authors consider including in Figure 3 an internal loop that was improved by the method, such as loop E or a kink-turn? While having two widely occurring, albeit simple, hairpins is useful, an internal loop example will also be helpful for demonstrating method's performance.

Answer:

Per suggestion, we have now added loopE results in Figure 3E. The best predicted conformation by Rosetta-SWM is about 2Å RMSD (panel E1).

However, these models have an incorrectly folded non-WC pair that prevented BRiQ to make significant further improvement over Rosetta-SWM (panel E2). If native non-WC pairs were employed, we would obtain 0.4 Å RMSD for the best within top 1% (panel E3). We can achieve this high-resolution structure even without using any non-WC pairs as restraints. This result is now described in the paper.

2) I wonder if the PDB structures that were part of the RNA Puzzles project have been excluded from the set used to calculate the BRiQ's parameters? If not, then it's conceivable that the improvement in the RNA Puzzles examples was due to the fact that the target structures have been seen by the method. As RNA Puzzles participants are likely to be the future users of BRiQ, it is important to show that BRiQ does well even for previously unknown 3D structures.

Answer: The homologous structures from RNA Puzzles contributed only to 0.1% of all structural data. We removed these structures from the statistics and found that the changes to the BRiQ energy score are negligible. Refinement results with the new BRiQ score are essentially the same (except those caused by stochastic nature of Monte Carlo sampling). This is now included in the discussion.

3) I downloaded the source code and tried installing it on my Mac, however, running "cmake ../" resulted in the following error:

```
BRiQ/build/CMakeFiles/FindOpenMP/OpenMPTryFlag.cpp:2:10: fatal error:  
'omp.h' file not found  
#include <omp.h>  
^~~~~~  
1 error generated
```

It could be useful to document all the requirements (for example, the installation instructions refer to cmake but it's not listed as a requirement).

Answer: A CMakeFile is made so that the program is now suitable to compile on a Mac machine.

The user documentation found in the Readme file is very limited and

should be expanded (currently it contains just 55 lines, including blank lines).

Answer: Additional information is now added.

4) The version of the software should be specified in the manuscript, and different versions of the software should be hosted separately to ensure the reproducibility of the work described in the paper. Did the authors consider hosting the source code on GitHub, BitBucket or similar platform to facilitate version tracking?

Answer: We now posted the final version of the code on <https://github.com/Jian-Zhan/RNA-BRiQ>

Typos

Should “Inter Xeon Gold” be “Intel Xeon Gold”?

Answer: Fixed.

Also, in addition to the CPU used for performance testing it will be useful to state the RAM requirements.

Answer: It is now stated that 2.9 G RAM was used during running.

Reference 43 does not appear to be formatted correctly.

Answer: Fixed. It is the reference for the Gaussian program, we took the citation format from their website.

Reviewer #2

1/ The authors derived the statistical energy terms from “2247 RNA structures obtained by X-ray crystallography and cryogenic electron microscopy with a resolution higher than 3.0Å downloaded on January 23, 2020”. I guess this was from the PDB, a reference is needed. Now the cut-off at 3.0 Å is too high. X-ray refinement at resolutions between 2.5 and 3.0 Å are not easy and contain many errors. There is no point of making highly sophisticated approximations and quantum-mechanical calculations on rather approximate structures with in-built errors. In this respect, the use of cryo EM structures is inappropriate since those used previous structures and are fitted in density and often with poor refinement (since only recently respectful refinement programs have been used). I realize they need lots of structures but adding noise (often in a redundant fashion) will not improve the parameters. They do state that they did not remove redundant sequences and, thus, several structures must be ribosomal structures, a lot of which are stereochemically full of errors because of the difficulties in data and the use of previous structures. There are only 1528 RNA structures present in the PDB now. So, I guess the authors used also RNP structures. What is the number of RNP structures? Are they all ribosomes? Do they include spliceosomes? In any case, one should only consider structures below 2.5 Å especially for feeding into quantum chemical programs. Do they use structures from the Cambridge data base? They are simpler structures but high resolution. The NDB contains also a lot of data on nucleic acids that are well curated.

Answer. Thank you for the question. We indeed obtained all structures from PDB (a reference is now added). We now included a statistics about the structural data (272 ribosome, 221 riboswitches, 138 tRNA, 106 ribozymes, 42 aptamers, 121 virus RNA, 17 introns, 3 spliceosomes and 1327 others). Among them, there are 1459 protein-RNA complex structures and 788 RNA-only structures. The base pairing information is indeed dominated by ribosome (about 89%). We did not use the Cambridge database because it contains simple structures only. To our knowledge, the structures in the NDB were included in the PDB. We clarify that quantum calculations were made on the highest resolution PDB structure within a cluster center of base-pairing structures and the initial structure from the PDB was further optimized quantum mechanically so as to minimize the effect of potentially inaccurate conformations. We agree that in a perfect world we would use all non-redundant structures in super high resolutions. If we used 2.5Å cutoff and X-ray structures only, we would only have 1225 structures with 10.5% bases of the current database only. This is too small for our purpose. We believe that the most important question is whether or not the BRiQ energy function can be applied to the unseen RNA structures. To test this, we removed RNAs in RNA Puzzles from the structural database and found that it leads to essentially the same BRiQ energy function and refinement results. Further discussion is added.

2/ What would be the results in terms of RMSDs if you submit to re-refinement some of the structures between 2.5 and 3.0 Å? Or those below 2.0 Å resolution (hoping some are present)?

Answer: As shown in Figure 4, BRiQ refinement always improves model structures with $\text{RMSD} < 3 \text{ \AA}$ in RNA puzzles and FARFAR2 sets, regardless which methods produced the models. We further performed the BRiQ refinement of native structures of RNA puzzles and examined the deviation from the native structure at the base-pair structural level. The new Supplementary Figure S2 shows the change of base pair structures in DDM as a function of X-ray structure resolution after refinement of native structures by BRiQ. Overall changes to the native base pairing structures are small. There is a trend that larger changes in base pair conformations were observed for lower resolution structures, suggesting more uncertainty for low resolution structures as expected. This is now described in the result section.

3/ Which dictionaries for bond lengths and angles were used? The dictionaries are not always standardized and thus, how variations in these fundamental geometrical parameters influence the final parameters and refinement?

Answer: We now clarify that the average coordinates were used for atoms in the bases. Atoms in riboses were generated from representative PDB conformations. Bond length and bond angles associated to the phosphate group were set to the average value in PDB library.

4/ “the interactions associated with bases and oxygen atoms in backbones are strongly orientation dependent.” Indeed true, but bond lengths and angles are also conformation dependent; this is clearly seen in ribose dimensions depending on the sugar pucker (see JACS 102, 1493 (1980)). Are these variations considered? Such a level of precision should be needed for quantum calculations.

Answer: This is a great question. We have added the reference to the paper and emphasized that atomic coordinates of riboses were not generated from a fixed bond length and angles but directly from the pucker-dependent conformers from PDB structures.

5/ MolProbity is used regularly for checking RNA structures. Is there an improvement in the clash score after re-refinement? For example, in some RNA Puzzles the clash

scores are high. Are there improved and if yes by how much? New Tables may be necessary (can be in sup mat).

Answer: We calculated MolProbity scores (Nucleic Acids Res 2007;**35(Web Server issue)**:W375-83) for all RNApuzzle structures. The new Supplementary Figure S1 shows that the clash scores of 75 or larger are all decreased to less than 50 after refinement. The average clash score reduced 40% from 20.87 to 12.58. Except a few outliers, the clash scores are less than 30 after refinement. This result is now included in the results section.

6/ Minor. In the abstract “the progress in protein or RNA structure refinement has been slow because native structures are often not the global minimum of existing approximate energy scores.” I think it should be stated the other way around: the global minimum given by the energy scores is not at the experimentally determined “native” structure.

Answer: Thank you. Changed.

Reviewers' Comments:

Reviewer #1:

Remarks to the Author:

The authors' response addressed most of my comments and the revised manuscript represents an improvement, although I still have some reservations about the use of all PDB structures for training without redundancy reduction. The authors argue that if the redundancy is reduced, then there is not enough data and since the algorithm seems to be working for non-rRNA structures, then all should be fine. I understand this logic but I still don't get how this additional information is obtained from the redundant structures, as they are often very similar, especially since reference structures are used as templates to model lots of other structures from the same species. I find the Supplementary Figure S3 rather cryptic despite the authors saying that "It is clear that the structures of same sequences can fill the conformational space missed by other structures, generating a more refined energy surface". However, if the method indeed works as described, maybe this overrepresentation of rRNAs does not have a negative effect.

I was able to install the code from the GitHub repo (<https://github.com/Jian-Zhan/RNA-BRiQ>), but when I tried running it using the example pdb file (gcaa.pdb) or another file downloaded from PDB with a 5S rRNA, I got an error, for example:

```
$BRiQ_Refinement demo/gcaa/gcaa.pdb output.pdb 123
rot lib
frag lib
ATOM 67 P G A 3 -3.029 -2.675 -7.881 1.00 0.00 P
ATOM 68 OP1 G A 3 -2.481 -3.278 -9.118 1.00 0.00 O
```

... (the rest of the file gcaa.pdb) ...

```
ATOM 326 H5 C A 10 -3.031 0.916 4.652 1.00 0.00 H
ATOM 327 H6 C A 10 -1.528 2.570 5.648 1.00 0.00 H
libc++abi.dylib: terminating with uncaught exception of type std::out_of_range: basic_string
Abort trap: 6
```

I am not sure if I am doing something wrong or if my Mac is not configured as expected by the BRiQ software.

In addition, the Readme file mentions a program BRiQ_init but it is not found under the build/bin folder after installation (the folder contains only BRiQ_Predict, BRiQ_Refinement, BRiQ_assignSS).

While the documentation has been expanded since the original version, it still appears to be insufficient (for example, it's unclear how one should test that the installation has been successful or what is the role of the \$RANDOMSEED command line argument). It seems to me that the software could benefit from a more extensive testing in order to be truly useful to a wide research community.

Typos

Line 311 "resolution structures, there are regions with have poor R-factor values or high clash scores."

Line 322 "not involved protein synthesis. Despite of this, BRiQ can consistently refine these noncoding" should be "not involved in protein synthesis. Despite of this, BRiQ can consistently refine these noncoding"

Reviewer #2:

Remarks to the Author:

I feel the authors answered constructively the questions and points I raised. It is important to pursue this type of research in innovative ways.

Eric Westhof

Reviewer #1 (Remarks to the Author):

The authors' response addressed most of my comments and the revised manuscript represents an improvement, although I still have some reservations about the use of all PDB structures for training without redundancy reduction. The authors argue that if the redundancy is reduced, then there is not enough data and since the algorithm seems to be working for non-rRNA structures, then all should be fine. I understand this logic but I still don't get how this additional information is obtained from the redundant structures, as they are often very similar, especially since reference structures are used as templates to model lots of other structures from the same species. I find the Supplementary Figure S3 rather cryptic despite the authors saying that "It is clear that the structures of same sequences can fill the conformational space missed by other structures, generating a more refined energy surface". However, if the method indeed works as described, maybe this overrepresentation of rRNAs does not have a negative effect.

Answer: Thank you. Indeed, there is no evidence to indicate otherwise.

I was able to install the code from the GitHub repo (<https://github.com/Jian-Zhan/RNA-BRiQ>), but when I tried running it using the example pdb file (gcaa.pdb) or another file downloaded from PDB with a 5S rRNA, I got an error, for example:

```
$BRiQ_Refinement demo/gcaa/gcaa.pdb output.pdb 123
```

```
rot lib
```

```
frag lib
```

```
ATOM 67 P G A 3 -3.029 -2.675 -7.881 1.00 0.00 P
```

```
ATOM 68 OP1 G A 3 -2.481 -3.278 -9.118 1.00 0.00 O
```

```
... (the rest of the file gcaa.pdb) ...
```

```
ATOM 326 H5 C A 10 -3.031 0.916 4.652 1.00 0.00 H
```

```
ATOM 327 H6 C A 10 -1.528 2.570 5.648 1.00 0.00 H
```

```
libc++abi.dylib: terminating with uncaught exception of type  
std::out_of_range: basic_string
```

Abort trap: 6

I am not sure if I am doing something wrong or if my Mac is not configured as expected by the BRiQ software.

Answer: The command “\$BRiQ_Refinement demo/gcaa/gcaa.pdb output.pdb 123” is incorrect. The first argument should not be a PDB structure file but an input file containing the location of the initial pdb file along with the base pair information and other restrains. We have modified our program to make a better input checking to avoid this type of unexpected crash.

In addition, the Readme file mentions a program BRiQ_init but it is not found under the build/bin folder after installation (the folder contains only BRiQ_Predict, BRiQ_Refinement, BRiQ_assignSS).

Answer: The Readme file was inadvertently misplaced. This is now corrected.

While the documentation has been expanded since the original version, it still appears to be insufficient (for example, it’s unclear how one should test that the installation has been successful or what is the role of the \$RANDOMSEED command line argument). It seems to me that the software could benefit from a more extensive testing in order to be truly useful to a wide research community.

Answer: \$RANDOMSEED is used to initialize the rand() function. If we run the refinement program twice with the same \$RANDOMSEED, we will get exactly the same results. We have now included three demo cases to illustrate three different usages along with additional comments in the README file and scripts. We plan to continuously update the documentation according to feedback from users.

Typos

Line 311 “resolution structures, there are regions with have poor R-factor values or high clash scores.”

Line 322 “not involved protein synthesis. Despite of this, BRiQ can consistently refine these noncoding” should be “not involved in protein synthesis. Despite of this, BRiQ can consistently refine these noncoding”

Answer: Thank you, corrected.

Reviewers' Comments:

Reviewer #1:

Remarks to the Author:

The expanded documentation enabled me to successfully run the software, which seems to be performing as described in the manuscript. I have no further suggestions and thank the authors for addressing all my comments.

Anton I. Petrov